

# Deuteration of proteins boosted by cell lysates: high-resolution amide and Hα MAS NMR without re-protonation bottleneck

Federico Napoli[1], Jia-Ying Guan[2], Charles-Adrien Arnaud[2], Pavel Macek[2], Hugo Fraga[2], Cécile Breyton[2], and Paul Schanda[1]

[1]Institute of Science and Technology Austria (ISTA), Am Campus 1, A-3400 Klosterneuburg, Austria
[2]Univ. Grenoble Alpes, CNRS, CEA, IBS, F-38000 Grenoble

**Correspondence:** Paul Schanda (paul.schanda@ist.ac.at)

**Abstract.** Amide-proton detected magic-angle spinning NMR of deuterated proteins has become a main technique in NMR-based structural biology. In standard deuteration protocols that rely on $D_2O$-based culture media, non-exchangeable amide sites remain deuterated, making these sites unobservable. Here we demonstrate that proteins produced with $H_2O$-based culture medium doped with deuterated cell lysate allow to overcome this "reprotonation bottleneck", while retaining a high level

of deuteration (ca. 80 %) and narrow line widths. We quantified coherence life times of several proteins prepared with this labelling pattern over a range of MAS frequencies (40-100 kHz). We demonstrate that under commonly used conditions (50-60 kHz MAS), amide $^1H$ line widths with our labelling approach are comparable to those of perdeuterated proteins and better than those of protonated samples at 100 kHz. For three proteins in the 33-50 kDa size range many previously unobserved amides become visible. We report how to prepare the deuterated cell lysate for our approach from fractions of perdeuterated cultures

which are usually discarded, and show that such media can be used identically to commercial media. The residual protonation of Hα sites allows for well-resolved Hα-detected spectra and Hα resonance assignment, exemplified by the *de novo* assignment of 168 Hα sites in a 39 kDa protein. The approach based on this $H_2O$/cell-lysate deuteration and MAS frequencies compatible with 1.3 or 1.9 mm rotors presents a strong sensitivity benefit over 0.7 mm/100 kHz MAS experiments.

## 1 Introduction

The deuteration of proteins for NMR studies is a very important technique and has greatly expanded the scope of biomolecular NMR. In solution-state NMR, overall Brownian motion modulates the relaxation-active interactions, in particular the strong $^1H$-$^1H$ dipolar interactions and chemical-shift anisotropy (CSA), and when this Brownian motion is slow, which is the case for large proteins, the resulting decay of coherences is fast. This rapid relaxation and the associated line broadening represent a fundamental barrier to the size of proteins that can be studied. It can be overcome by using highly deuterated proteins, where

only a few sites, e.g. amides, methyls or aromatic sites, bear a $^1H$ spin, while all the rest is deuterated ($^2H$). Together with transverse-relaxation optimised pulse sequences tailored for amides Pervushin et al. (1997), methyls Tugarinov et al. (2003) or aromatics Pervushin et al. (1998), deuteration has alleviated the size limitations of solution-state NMR, and has made proteins in the megadalton range accessible to site-resolved studies of dynamics and interactions.



In magic-angle spinning (MAS) solid-state NMR, overall molecular tumbling, that dominates relaxation in solution is absent,
and the transverse relaxation is inherently slow. However, the dipole-dipole interactions are not perfectly averaged by magic-angle spinning, and these interactions lead to a rapid decay of the observable spin coherences. Traditionally, biomolecular MAS NMR has, thus, focused on observing $^{13}$C rather than $^1$H. The smaller gyromagnetic ratio of $^{13}$C and accordingly the smaller dipolar couplings lead to a more complete averaging of the dipolar couplings already at lower MAS frequency. However, it also comes with inherently lower detection sensitivity than $^1$H. In order to achieve narrow $^1$H line widths, diluting the network of $^1$H-$^1$H dipolar interactions by deuteration is a well established method. Deuteration with sparse introduction of $^1$H nuclei only at e.g. amide ($^1$H$^N$), Hα, methyl or aromatic sites, has rapidly established itself as a standard technique in biomolecular MAS NMR. The higher detection sensitivity of $^1$H translates to smaller sample amounts needed. Pioneering experiments e.g. by the Reif group used perdeuterated proteins in which only a fraction (e.g. 10 %) of the amides are protonated, thereby sacrificing signal intensity, at MAS frequencies of 20 kHz, i.e. with 3.2 mm large rotors and ca. 30 mg material Chevelkov et al. (2006). Faster MAS with smaller-diameter rotor allows to introduce more protons while keeping narrow lines; 100% amide-protonation with MAS frequencies of 40-60 kHz, i.e. smaller diameter rotors of either 1.3 mm (ca. 2.5 mg material) or 1.9 mm (ca. 13 mg) has become of widespread use Lewandowski et al. (2011); Zhou et al. (2007). Akin to solution-state NMR, amide-$^1$H-detected MAS NMR experiments are employed for many aspects of protein structure and dynamics studies: they are used to obtain backbone resonance assignment Barbet-Massin et al. (2014); Fraga et al. (2017); Xiang et al. (2015); Stanek et al. (2020); Andreas et al. (2015); Klein et al. (2022), determine structures from $^1$H$^N$–$^1$H$^N$ distances Retel et al. (2017); Zhou et al. (2007); Knight et al. (2012); Linser et al. (2011); Najbauer et al. (2022); Gauto et al. (2019a); Schubeis et al. (2020) or long-distance restraints to paramagnetic Knight et al. (2012) or $^{19}$F Shcherbakov et al. (2019) sites, or to characterise protein dynamics Napoli et al. (2023); Singh et al. (2019); Bonaccorsi et al. (2021); Good et al. (2014); Lewandowski (2013); Chevelkov et al. (2009) (We refer the reader to excellent recent reviews of $^1$H-detected MAS NMR and deuteration Le Marchand et al. (2022); Vasa et al. (2018); Reif (2021).) Very high MAS frequencies available today with 0.5 mm or 0.7 mm rotors (up to 150 kHz) alleviate the need for deuteration, and protonated proteins are sufficient for structure determination or dynamics studies Agarwal et al. (2014); Andreas et al. (2016); Lamley et al. (2014); Bougault et al. (2019); Cala-De Paepe et al. (2017). However the absolute detection sensitivity drops substantially due to the smaller rotor volume; for example, using a 0.7 mm rotor instead of a 1.3 mm rotor, each close to its respective maximum MAS frequency, leads to a loss of at least a factor of 2.5 in sensitivity (or 6 in experimental time) Le Marchand et al. (2022). Thus, for many practical applications, the use of deuterated proteins with MAS frequencies in the 40-60 kHz range (1.9 or 1.3 mm rotors) is still a preferable method.

Deuterated proteins for amide-detected experiments are commonly produced by bacterial overexpression in a minimal growth medium Gardner and Kay (1998) (M9 medium Anderson (1946)) prepared with $D_2O$. In the following we refer to this method of deuteration in $D_2O$ based growth media as perdeuteration. As the amide hydrogen atoms are chemically exchangeable, those accessible to the solvent are replaced with the $^1$H isotope when the perdeuterated protein is placed in $H_2O$ during purification and subsequent measurement. This amide re-protonation is, however, often incomplete, in particular in the core of large proteins, which leads to lack of information for these parts of the structure. As a consequence of the incomplete



back-exchange of amide sites, many structurally important probes are invisible. For example, in the structure determination of TET2 Gauto et al. (2019a), for a buried β sheet only very few distance constraints were available.

One option to circumvent the need for re-protonation of amide sites is to focus on Hα sites instead. An elegant approach has been proposed by the Andreas group, which starts from a commercial growth medium, which contains a mix of deuterated amino acids, and makes use of transaminases to protonate the Hα positions Tekwani et al. (2019), in an amino-acid-type dependent manner. Over-expression is then performed with this mix of deuterated Hα-protonated amino acids. The method, named "alpha proton exchange by transamination" (α-PET), has been shown to generate proteins that are protonated predominantly at

the α sites. A potential drawback is, however, that Hα sites generally have narrower signal dispersion than amide sites, such that resolution is often lower. Moreover, dynamics studies using the Cα site are more complicated, because the $^{13}$C-$^{13}$C couplings render any quantitative analysis of relaxation complex (if not impossible); dynamics studies using $^{15}$N sites are preferable.

Denaturation of the perdeuterated protein in $H_2O$, followed by refolding is one viable route to achieve complete re-protonation of amide sites in perdeuterated proteins Gardner et al. (1998). However, it is often difficult or even impossible to find suitable

conditions in which the protein retrieves its native structure.

An alternative way to obtain (at least partly) deuterated proteins in which all exchangeable hydrogen sites bear a $^1$H isotope is to perform the deuteration in $H_2O$, while providing the building blocks that serve for protein synthesis in a deuterated form. A simple approach is to use deuterated D-glucose as sole carbon source in $H_2O$-based minimal (M9) medium; this approach has been proposed for MAS NMR and termed inverse fractional deuteration (iFD) Medeiros-Silva et al. (2016). As the entire

biosynthesis of amino acids from glucose takes place in $H_2O$, this approach results in relatively low deuteration levels; in particular, the α sites have been reported to be protonated to 88-100 % for all reported amino acids (Table S2 of Medeiros-Silva et al. (2016)). Due to the low overall deuteration the coherence life times of the amide $^1$H spins in the iFD scheme are close to those of protonated samples, i.e. the gain in resolution with this sparse deuteration is limited Cala-De Paepe et al. (2017).

Using deuterated amino acids in the growth medium promises to result in more complete deuteration. Löhr *et al.* grew

cultures in $H_2O$-based medium with a commercial deuterated algal lysate (without adding glucose) for solution-state NMR study of a 35 kDa-large protein. This approach has not been reported for MAS NMR, to our knowledge. A significant drawback of this method is the price. The lysate that Löhr *et al.* used has a current list price of US$ 6500 for 1 litre of culture (Dec 2023), which may be considered prohibitively expensive in many cases. Moreover, the amino acids are partly re-protonated due to the action of transaminases, and this protonation largely varies depending on the amino acid type; the residual protonation at

α sites can reach 80 % Löhr et al. (2003). Besides bacterial expression, cell-free protein production in $H_2O$ with deuterated amino acids can be performed. When used with transaminase inhibitors, the level of α protonation can be maintained below ca. 10 % Imbert et al. (2021); Xuncheng et al. (2011). However, few laboratories have established cell-free production, and this approach is less readily accessible than bacterial over-expression.

Here, we propose an approach for producing highly (≥ 80 %) deuterated proteins in $H_2O$-based M9 medium with deuterated

D-glucose supplemented with deuterated cell lysate. We find that 2 g of cell lysate powder added to the medium suffice to reach 80 % overall deuteration level, and this level is ca. 4 times higher than the one obtained with the previously proposed iFD approach Medeiros-Silva et al. (2016). While commercial deuterated media can be used, we report the straightforward



preparation of a home-made cell lysate from what is usually discarded from perdeuterated bacterial protein production, and show that its properties are indistinguishable from a commercial medium.

Moreover, we investigate the $^1$H coherence life times of several proteins produced with this $H_2O$/M9/cell-lysate deuteration scheme, and compare them to those of perdeuterated and fully protonated protein samples over a range of MAS frequencies up to 100 kHz. Most importantly, we report that the amide $^1$H line widths of the $H_2O$/M9/cell-lysate deuterated samples are not significantly different from those of perdeuterated samples (although the coherence life times, i.e. the homogeneous line widthsLe Marchand et al. (2022), of the perdeuterated samples are favorable). Using three proteins with molecular weights ranging from 33 to 50 kDa in 1.3 mm or 1.9 mm rotors (38-55 kHz MAS frequency), we show that our approach retrieves many amide signals that were lost in a standard perdeuteration approach. For the 12 x 39 kDa large protein assembly TET2, we obtained 40 % more amide assignments than with the previous perdeuteration approach, most of which in the hydrophobic core of the protein. We show furthermore that the residual "unwanted" protonation of α sites can be exploited in 3D and 4D experiments that edit the Hα frequency. We obtained 168 Hα chemical-shift assignments, which is close to the number obtained from a fully protonated sample at 100 kHz MAS.

## 2  Results

### 2.1  Deuteration of proteins in $H_2O$-based cultures with deuterated algal lysate

The deuteration level and deuteration pattern (i.e. which sites are deuterated) is essential for the resolution in amide-hydrogen based NMR MAS spectra. We first quantified the overall deuteration level that can be reached using $H_2O$-based bacterial cultures producing the 33.55 kDa protein MalDH, by determining the intact protein mass with mass spectrometry (Figure 1A). When only deuterated glucose is used (at 2 g per litre of culture; this corresponds to the previously proposed "iFD" approach Medeiros-Silva et al. (2016)), the resulting protein is deuterated to ca. 25% overall. The addition of deuterated algal lysate-derived complex labeling medium increases the deuteration level in a concentration-dependent manner: upon addition of ca. 2 g/L of ISOGRO® powder to the medium, a plateau level of ca. 80% deuteration is reached. Doubling the amount of ISOGRO® added to the medium to 4 g/L has only very small effect, such that we consider 2 g/L as a sufficient amount. For brevity, we refer in the remainder of this text to the deuteration that uses $H_2O$ culture medium with deuterated M9 components (including 2 g/L deuterated D-glucose) and deuterated cell lysate (at 2 g/L) as "$H_2O$/M9/lysate" sample. With lysate we refer either to the commercial complex growth medium (such as ISOGRO®) or a in-house made variant of it (see below). We refer to "perdeuteration" as the approach using $D_2O$-based M9 medium using 2 g/L deuterated D-glucose. In all cases, the final NMR samples are in a buffer composed of $H_2O$.

Having quantified the overall deuteration level, we investigated more closely the pattern of the residual protonation. To this end, we have produced the 8.6 kDa protein ubiquitin with different labelling patterns: either no deuteration (u-[$^{13}$C,$^{15}$N]), perdeuteration (u-[$^2$H,$^{13}$C,$^{15}$N]), or deuteration with the $H_2O$/M9/lysate approach proposed herein. For the latter, the M9 culture medium was in $H_2O$ and included 2 g/L $^2$H,$^{13}$C glucose, 1 g/L $^{15}$N ammonium and 2 g/L $^2$H,$^{13}$C,$^{15}$N ISOGRO® powder.





125 We measured the deuteration level with $^1$H-$^{13}$C HSQC spectra, which provides simultaneously the deuteration levels for all aliphatic sites.

Figure 1B and C show the protonation levels in aliphatic side chains and at Hα positions. Marked differences were found for Hα deuteration of different amino acid types, qualitatively similarly to previously reported data Löhr et al. (2003). For example, while Glu and Phe have a high incorporation of $^1$Hα, Lys and Ala have a higher deuteration level. Positions further 130 out in the side chain have a higher deuteration level. Overall, our data show that there is significant variation in the deuteration level, which we ascribe to the activity of transaminases that differs for the types of amino acids. While the residual protonation may also be of possible use, namely for $^1$H detection of aliphatic hydrogen sites, as we will show below, the presence of these $^1$H spins in vicinity to amide protons will likely accelerate the coherence decay of amide $^1$H spins. This question of coherence life times and line widths is addressed in a later section in this manuscript.

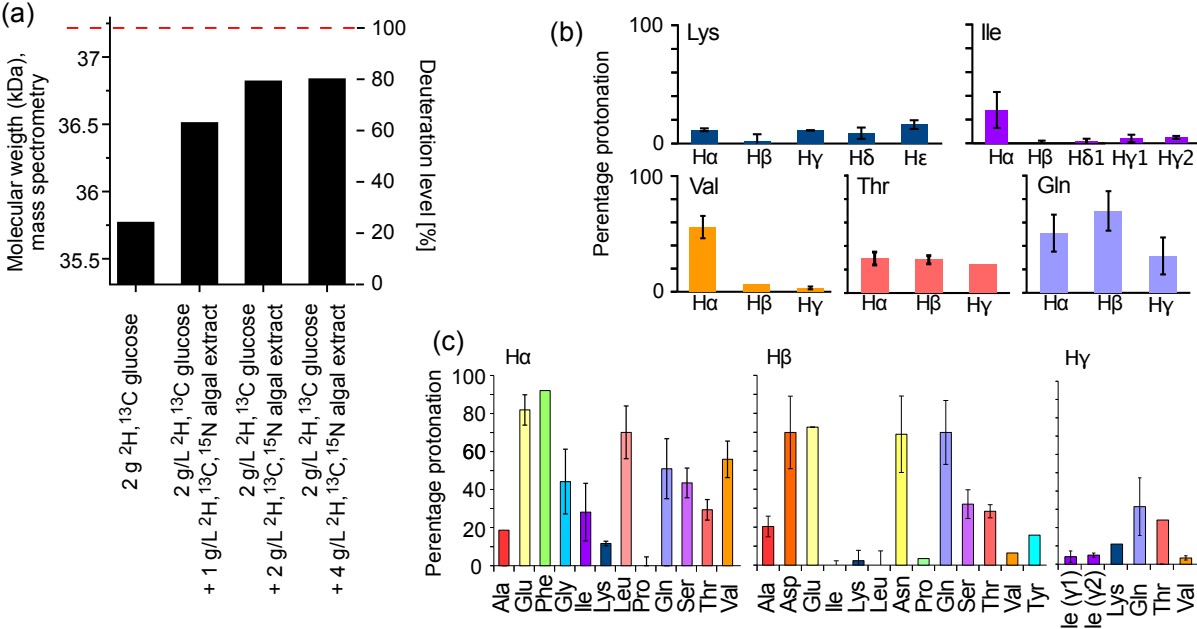

**Figure 1. Deuteration using H$_2$O-based medium doped with algal lysate.** (A) Deuteration level of *I. islandicus* MalDH (33.55 kDa) expressed in H$_2$O with either $^2$H,$^{13}$C glucose only (2 g per liter of culture) and $^{15}$NH$_4$, or with additional use of $^2$H,$^{13}$C,$^{15}$N algal extract (ISOGRO®) at three different concentrations (1, 2, 4 g per liter of culture). The reported molecular-weight values are from intact mass spectrometry. The dashed red line indicates the theoretical molecular weight of fully deuterated MalDH (assuming all exchangeable hydrogens are $^1$H). (B,C) Residual protonation level for aliphatic sites, determined by solution-state NMR of a sample of ubiquitin produced in H$_2$O-based M9 medium supplemented with $^2$H,$^{13}$C,$^{15}$N algal extract (2 g/L).

## 2.2 Straightforward preparation of an isotope-labelled cell lysate as an alternative to commercial media

Having shown that a commercial deuterated algal extract (e.g. ISOGRO® from Sigma-Aldrich or Celtone® from Cambridge Isotope Laboratories or SILEX® media from Silantes) allows obtaining high deuteration levels in $H_2O$-based bacterial expression, we have investigated the possibility to prepare such a medium in house. In many laboratories, deuterated proteins are prepared from *E. coli* cultures grown in $D_2O$ based M9 media. While the $D_2O$ from such cultures is often reused, after distillation, the contaminant proteins and the insoluble pellet are usually discarded. We have set up a very simple method to make use of this deuterated material, and use it in the same way as the commercial lysate to deuterate proteins.

In principle, different parts may be "recycled" after the protein of interest is purified: either the soluble contaminant proteins or the insoluble debris. We have explored both possibilities. The soluble contaminant proteins were retained during the Ni-affinity purification step; the insoluble fraction was simply the part that was in the pellet after a centrifugation following cell lysis. In both cases, the sample was treated with phosphoric acid to hydrolyse the peptide bonds. The pH was neutralised by NaOH and the sample was cleared by ultracentrifugation and then lyophilised. We have obtained ca. 375 mg of lyophilised

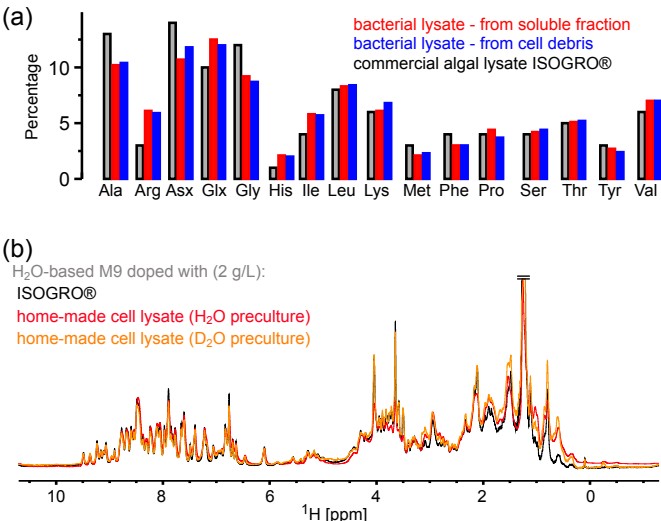

**Figure 2. Comparison of the amino-acid composition in a home-made bacterial extract with the one in ISOGRO®.** (A) The amino acid composition of ISOGRO®, as provided by the manufacturer, is shown as grey bars; the values represent the percentage (weight) of each amino acid in the powder. Data of the home-made preparation from bacterial perdeuterated cultures are shown in blue and red. The preparation made from the soluble-proteins fraction, i.e. contaminant proteins from a protein purification, is shown in red. The preparation made from the insoluble fraction after cell lysis is shown in blue. Asn and Gln are indistinguishable from Asp and Glu with this method. Cys and Trp are destroyed during the processing steps of the analysis. (B) 1D [1]H spectra of ubiquitin deuterated in $H_2O$ based medium supplemented with either commercial algal lysate (black) or the home-made lysate, prepared from the contaminant proteins. The precultures have been made either in $H_2O$ or $D_2O$, as indicated. The similarity of the spectra confirms that commercial algal lysates and bacterial lysates produce similar labelling patterns, as expected from the very similar distribution of amino acids shown in panel (A).



powder from 1 L of culture using the soluble fraction and ca. 900 and 1400 mg (two independent samples) from the insoluble debris. We have analysed the amino-acid composition of these powders by amino-acid analysis. Figure 2 compares the relative amino-acid composition of these two samples to the composition of the commercial ISOGRO® medium. The composition is
similar, which suggests that the cell lysate from bacteria can be used in the same manner as the commercial algal lysate.

To test this assumption, we have produced samples of ubiquitin with either deuterated commercial algal lysate (ISOGRO®) or the home-made bacterial lysate described above (2 g/L each), and recorded NMR spectra (Fig. 2B). The spectra show a very similar level of deuteration, supporting that bacterial cell lysates is a viable alternative to commercial ones. The bacterial cell lysate comes essentially free of additional cost when perdeuterated proteins are made, considering that it uses fractions that are
usually discarded. However, with a yield of ca. 1 g per litre of culture, ca. 2 L of a deuterated culture are needed to prepare 1 L of $H_2O$/M9/cell-lysate-deuterated sample with good deuteration (cf. Figure 1A).

### 2.3   Coherence life times in $H_2O$/M9/cell-lysate deuterated samples

Two key parameters we are interested in are the coherence life times ($T_2$') and the line widths. The line width contains contributions from the coherence life time (the so-called homogeneous contribution), but additional sample inhomogeneity
also contributes Le Marchand et al. (2022). Which of these contributions dominates depends on the sample and the conditions (in particular the MAS frequency). To compare the $H_2O$/M9/cell-lysate samples with perdeuterated and protonated samples, we have, thus, analysed both the homogeneous contribution and the apparent line widths, in two large test proteins, TET2 (39 kDa monomer size, assembling to a dodecamer) and MalDH (33.55 kDa monomer size, assembling to a tetramer). To quantify the homogeneous contribution we have measured the coherence decay using spin-echo experiments. The experiments applied
a spin echo ($\tau$ – 180° pulse – $\tau$) element to either $^1H$, $^{15}N$ or $^{13}CO$ coherence, embedded into a heteronuclear correlation experiment with $^1H$ detection. We compared samples prepared with full deuteration (produced in $D_2O$ medium) with samples deuterated in $H_2O$ medium doped with deuterated algal lysate, and finally a fully protonated sample. As shown above, these three samples correspond to different levels of deuteration. In all cases, the proteins were in $H_2O$ buffer, i.e. all the exchangeable sites are protonated to either 100% (for the two types of samples grown in $H_2O$ medium) or to a level that depends on the
accessibility of the given amide site. Because the importance of deuteration depends on the MAS frequency Le Marchand et al. (2022), we have performed the measurements at various MAS frequencies up to 100 kHz (Figure 3).

As expected from the deuteration level, the $^1H$ coherence life times of the samples deuterated with algal lysate in $H_2O$ (red in Figure 3) are in between those of perdeuterated samples (blue) and the fully protonated sample (black). Of note, even at the highest MAS frequency used in this study, 100 kHz, the $^1H$ coherence life time of the fully protonated sample is more than
three times shorter than the one of the sample deuterated with algal medium. Moreover, the $^1H$ coherence life time of the fully protonated sample spinning at 100 kHz is significantly (more than 20%) shorter than the one of the algal-medium-deuterated sample spinning at 60 kHz. This is an important realisation, because in order to be able to spin 100 kHz, one needs to use smaller rotors and sacrifice about ca. a factor of 5 in sample amount, which translates to a severe sensitivity penalty of 100-kHz MAS experiments. As expected from the deuteration level study above, the perdeuterated sample (placed in $H_2O$) has longer





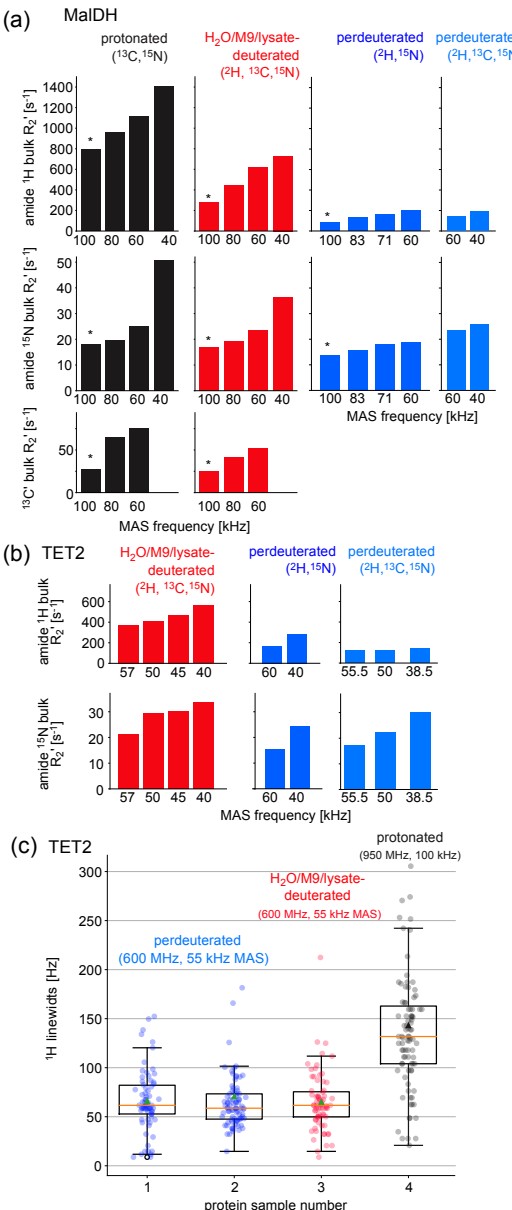

**Figure 3. Investigation of coherence life times and line widths in differently labelled samples of two proteins.** (A) Fitted $R_2'$ decay rate constants for four differently labelled samples of MalDH as indicated. The decay curves for representative experiments, indicated with an asterisk, are shown in Figure A1. All data were recorded at a $B_0$ field strength corresponding to 700 MHz $^1$H Larmor frequency in a 0.7 mm probe. (B) Similar data for the TET2 protein. (C) $^1$H line widths of TET2 in deuterated samples produced in $D_2$O-based M9 medium (blue), produced in $H_2$O-based M9 medium doped with 2 g/L deuterated algal extract (red) or fully protonated (black). Spectra are shown in Fig. A2.



[1]H coherence life times (lower $R_2$'). Moreover, in absolute terms, the MAS dependency is less pronounced for the deuterated sample than the protonated one.

The decay of heteronuclear (amide [15]N, carbonyl [13]C') coherences depends much less on the deuteration level (Figure 3). It depends significantly on the [1]H decoupling scheme and decoupling power used, and the optimum decoupling scheme and decoupling power generally depends on the MAS frequency. To generate a consistent data set, we have used low-power [1]H

WALTZ-16Shaka et al. (1983) decoupling (ca. 10 kHz), which is not the best-performing scheme at 40 kHz (data not shown). Therefore, these data are to be taken with a grain of salt, and the main message from them is that [15]N and [13]C' decay is much less dependent on the labelling pattern than [1]H coherence life times. Of note, the experiments performed here did not use [2]H decoupling; the rationale is that most probes are not equipped with a [2]H coil.

## 2.4 Line widths in H$_2$O/M9/cell-lysate deuterated samples at 55 kHz MAS are similar to perdeuteration and better
than protonated samples at 100 kHz

The line widths observed in spectra contain additional inhomogeneous contributions due to sample heterogeneity, inhomogeneity of the magnetic field and anisotropic bulk magnetic susceptibility Le Marchand et al. (2022); Linser et al. (2014). The $R_2$' rate constants discussed above do not comprise these effects. We have measured the line widths in a series of 2D hNH spectra obtained with TET2 produced either with the H$_2$O/M9/lysate approach or by perdeuteration (Figures 3C and A2). We

find that the amide [1]H line widths in samples deuterated in H$_2$O with deuterated algal lysate are comparable to those of proteins produced in D$_2$O. Importantly, the line widths of deuterated proteins at 55 kHz MAS frequency are significantly smaller than line widths of the same protein in protonated form, spinning at 100 kHz MAS frequency.

Taken together, the above analyses showed that proteins deuterated in H$_2$O with deuterated cell lysate, and 100% back-exchanged in the final sample, have slightly worse spectroscopic properties (shorter [1]H coherence life times) than perdeuterated

proteins under comparable conditions; the [1]H line widths, however, are not significantly larger.

## 2.5 Application to three large proteins: retrieving the signals of non-exchangeable amide sites

To show the potential of this deuteration of proteins with full amide-protonation, we have recorded three-dimensional correlation spectra of three large proteins, TET2, MalDH and the 50 kDa-large bacteriophage T5 tail protein pb6. Figure 4 shows the overlays of the hNH, hCONH and hCANH spectra of TET2, and hCANH spectra of MalDH and pb6. It is apparent in

all three cases that numerous peaks which are undetected in the standard perdeuteration protocol (blue) are visible in the H$_2$O/M9/lysate-deuterated protein (red).

To gain more insight, we have analysed in detail which amide signals are visible in TET2 with the two different deuteration approaches. TET2 is an ideal real-world application. With 39 kDa monomer size, it is one of the largest proteins for which extensive assignments have been reported Gauto et al. (2019a). The assignment of TET2 has been achieved with a

combination of 3D and 4D [13]C-detected experiments on protonated samples, and [1]H-detected 3D correlation experiments on perdeuterated TET2. 85% of the backbone and 70% of the side-chain heavy atoms have been assigned in this way. 140 amide hydrogen frequencies have been assigned using the standard perdeuterated samples (see Table A1). TET2 is a very stable

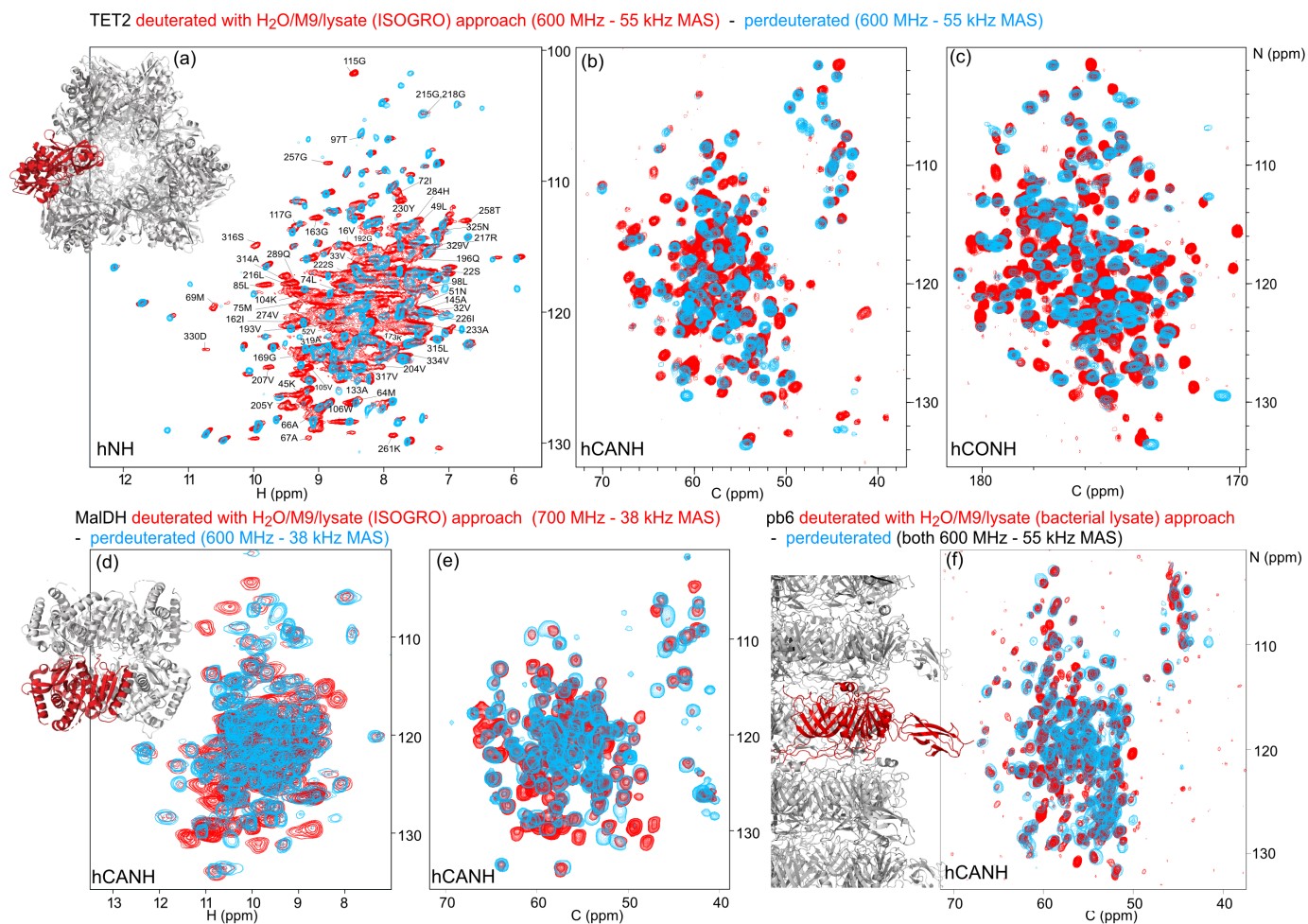

**Figure 4. Deuteration in H$_2$O enhanced by amino-acid mixtures from cell lysates allows detection of non-exchangeable amide hydrogens.** Overlay of (A) hNH, (B) hCANH, (C) hCONH TET2 spectra, (D) H-N and (E) CA-N projections of the hCANH MalDH spectra, and (F) hCANH pb6 spectra. (For the latter, the in-house made bacterial cell lysate was used.) Spectra of samples deuterated by the H$_2$O/M9/lysate approach are showed in red, while spectra from perdeuterated samples are shown in blue. All samples were $^{13}$C and $^{15}$N labelled. Peaks that appear to only be present in the perdeuterated sample spectra can be explained by (i) different apparent frequencies of aliased peaks (since the spectral widths and carriers are not identical in the overlayed spectra), (ii) chemical shift perturbations due to small sample-temperature differences, and (iii) different signal/noise ratios of individual peaks.

protein, originating from a hyperthermophilic archaeon, which suggests that the protein rarely populates partially unfolded conformations that would be required to re-protonate the amides. Unfolding/refolding comes with large sample losses, such that the H$_2$O/M9/lysate approach is ideal.






We have recorded 3D assignment experiments (hCANH, hCONH) with a sample deuterated in $H_2O$ with cell lysate. These experiments allowed us to assign an additional 58 amide resonances (see Table A1). Figure 5 plots the location of these newly assigned amide resonances in the structure and shows that the majority of them are located in the β-sheets of TET2 as well as in several α-helices. Interestingly, also a few resonances in more exposed regions had not been assigned before and have now

become assigned.

With a total of ca. 200 assigned amide hydrogens for a protein of 338 non-proline residues, there are still missing assignments left. This is in part explained by the fact that ca. 15% of the backbone heavy-atom resonances have not been assigned by [13]C-detected experiments, which is ascribed at least for some parts to extensively fast transverse relaxation due to μs dynamics Gauto et al. (2022). Moreover, the somewhat limited set of 3D experiments recorded here were not sufficient to unambiguously

assign all detected spin systems, as the intention of this study was to investigate the labelling scheme rather than to do an extensive resonance-assignment effort.

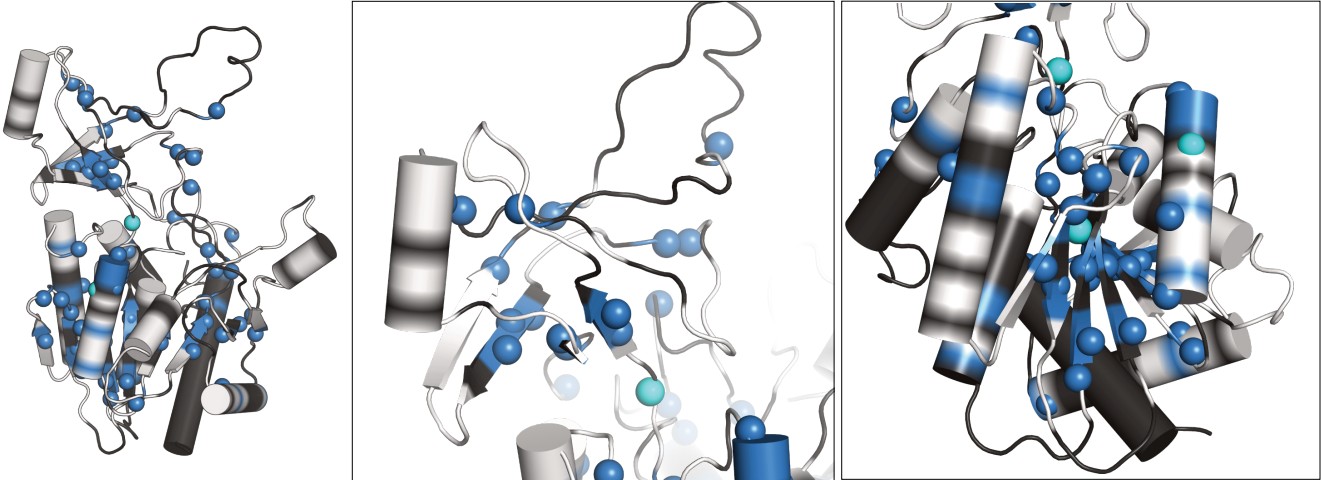

**Figure 5. Newly assigned amide hydrogens in TET2 deuterated in $H_2O$ shown on its monomeric structure.** Amide [1]H frequencies that were previously inaccessible with perdeuterated samples and that were assigned with the $H_2O$/M9/lysate deuteration are represented as spheres on the protein structure. Blue spheres correspond to atoms that could be assigned from the spectra of the $H_2O$/M9/lysate-deuterated sample (600 MHz [1]H Larmor frequency, 55 kHz MAS), while cyan spheres are for atoms that only showed signal in the spectra of the fully protonated sample (950 MHz [1]H Larmor frequency, 100 kHz MAS). Parts for which the backbone is shown in white regions correspond to those for which heavy-atom assignments have previously been reported Gauto et al. (2019a), while black regions indicate parts with missing heavy-atom assignment. In those parts we did not attempt to get new amide assignments, as the *de novo* assignment of the backbone would likely require more 3D data sets. Taken together, this data highlights the possibility of detecting amide hydrogens of water-inaccessible regions in the protein. Note that the protein forms a dodecamer (Fig. 4A), and for better visibility only the monomeric subunit is shown.



We investigated whether an alternative approach, namely the use of fully protonated samples spinning at 100 kHz would allow the assignment of many additional resonances: hCANH and hCONH experiments allowed us to assign only three additional amide frequencies, which brings the number of assigned amide [1]H sites to 201 (Table A1). We note that the sensitivity of these experiments, recorded in 0.7 mm rotors, is much lower than the one of experiments in 1.3 mm rotors, such that the use of samples deuterated in $H_2O$ represents a clear sensitivity advantage.

Overall, these data exemplify how the use of proteins deuterated in $H_2O$ provides access to amide sites that are difficult to exchange, e.g. buried in the hydrophobic core.

## 2.6 Residual Hα protonation enables Hα-based experiments and Hα assignment

Our investigation of the deuteration pattern (Figure 1) and, similarly, previously reported results Löhr et al. (2003) show that there is a significant fraction of the α sites which bear a [1]H spin. While this residual protonation reduces the coherence life times of amide [1]H spins, it can also be turned to an advantage, as the [1]Hα frequency becomes available as an additional reporter.

We have exploited the residual [1]Hα protonation in the $H_2O$/M9/lysate-deuterated sample to assign [1]Hα frequencies in TET2. To this end we have recorded a 4D Hα-Cα-N-H correlation experiment with three cross-polarisation steps (HACANH), at 55 kHz MAS frequency. As the Cα, N and H frequencies are known for most residues, the assignment of the Hα frequency is straightforward with this experiment. Figure 6A shows examples from this 4D experiment. This experiment allowed the assignment of 157 Hα resonances (Table A1).

In addition, we have recorded a complementary assignment experiment with a [13]C, [15]N labelled (protonated) sample at 100 kHz MAS frequency, which correlates the Hα to the directly bonded [13]Cα and the adjacent [15]N (hNCAHA; Figure 6B, grey). These two experiments both report on the Hα resonances. The 3D hNCAHA experiment, also based on three cross-polarisation steps, relies on the ability to resolve [13]Cα-[15]N correlations in two dimensions, in order to unambiguously connect the Hα frequency, while the 4D HACANH spreads the signal across three previously assigned dimensions and, therefore, is better in providing unambiguous assignments. 11 additional Hα assignments were obtained from the experiment on the protonated samples. The assigned Hα-Cα resonances are indicated on the 2D hCH spectrum in Figure 6C, and the location of the Hα assignments along the sequence is shown in Figure 6D and E. Hα assignments provide additional secondary-structure information and may help to better define the secondary-structure assignments using programs such as TALOS-N Shen and Bax (2013); the comparison of TALOS-N results with and without the Hα assignments (Fig. A4) shows only small differences.

Taken together, while the partial protonation of α sites is in principle an unwanted by-product of the deuteration in $H_2O$, we have shown here that the Hα nuclei can be used as an additional nucleus with good resolution, already at MAS frequency of 50-60 kHz, where fully protonated samples show poorly resolved aliphatic [1]H peaks.





**Figure 6. Protein deuteration in H₂O allows assignment of Hα hydrogens**



**Figure 6.** (continued) (A) Example strips from 4D HACANH and (B) 3D hNCAHA, hCANH correlation experiments exploited for the assignment of Hα hydrogens. (C) Overlay of the Hα region of the hCH spectra for the fully protonated (grey) and deuterated in $H_2O$ (red) samples. The same color scheme applies to panels A and B. (D) Newly assigned Hα hydrogens are represented as spheres on the protein structure. Green spheres correspond to atoms that could be assigned from the spectra of the sample deuterated in $H_2O$, while orange spheres are for atoms that only showed signal in the fully protonated sample's spectra. Black regions indicate missing Hα assignment. (E) Sequence-based representation of TET2 resonance assignments. Newly assigned amide and Hα hydrogens are shown in blue and green, respectively. Lost assignments correspond to sites for which the automated-assignment software FLYA Schmidt and Güntert (2012) did not converge after adding the peak list from the $H_2O$/M9/lysate-deuterated sample to the ones used previously for automatic assignment Gauto et al. (2019a). Protein deuteration in $H_2O$ allows assignment of Hα hydrogens.

## 3 Methods

### 3.1 Protein samples

*P. horikoshii* TET2 production was achieved by over-expressing pET-41c plasmid in *E. coli* BL21(DE3) competent cells (Novagen). The plasmid is available at AddGene (deposition number 182428). Culture media based on M9 minimal media were used for the different labeling schemes: (i) u-[$^2$H,$^{13}$C,$^{15}$N] labelled (perdeuterated) samples were expressed in M9 minimal media, 99.8% $D_2O$, $^{15}$NH$_4$Cl as sole nitrogen source, and deuterated and $^{13}$C$_6$-labeled glucose; (ii) $H_2O$/M9/lysate-deuterated samples were expressed in $H_2O$ M9 minimal media, supplemented with u-[$^2$H,$^{13}$C,$^{15}$N] labeled algal lysates (2 g/L of culture unless specified otherwise; Sigma-Aldrich) and $^2$H,$^{13}$C D-glucose as an additional carbon and energy source. (iii) u-[$^1$H,$^{13}$C,$^{15}$N] labeled (protonated) samples were expressed in $H_2O$ M9 minimal media, integrating $^{15}$NH$_4$Cl and protonated $^{13}$C$_6$ D-glucose. Cells were grown at 37 °C until OD (600 nm) reached ca. 0.6-0.8. Protein expression was induced using isopropyl-β-D-1-thiogalactopyranoside (IPTG; 1 mM). Cells were harvested by centrifugation and pellets were resuspended in Lysis buffer T (Table A2), and disrupted using a Microfluidizer. Cell lysates were heated to 85 °C for 15 minutes and subsequently centrifuged (17500 rcf, 1 hour, 4 °C). The supernatant was dialyzed overnight against Dialysis buffer at room temperature and re-centrifuged as in the previous step. Protein purification was conducted using a Resource Q column (GE Healthcare), eluting TET2 using a linear gradient of Elution buffer over 10 column volumes. Fractions were analysed by SDS-PAGE (12.5 % polyacrylamide) and TET2 was identified by its monomeric molecular weight (39 kDa) and concentrated using an Amicon 30 kDa concentrator (Millipore). The concentrated solution was loaded onto a HiLoad 16/60 Superdex 200 column (GE Healthcare) equilibrated with Dialysis buffer. Samples for MAS NMR measurements were prepared as described earlier Gauto et al. (2019a), by concentrating TET2 to 10 mg/mL in NMR buffer. The solution was then mixed (1:1 vol/vol) with 2-methyl-2,4-pentanediol (MPD). Protein precipitates were packed into 0.7/1.3/1.6/3.2 mm MAS rotors by ultrecentrifugation (50000 g, at least 1 hour).

*I. islandicus* malate dehydrogenase was produced by over-expression of pET-21a(+) plasmid in *E. coli* BL21(DE3) competent cells (Novagen). M9 minimal-based media were used as for TET2; $H_2O$/M9/lysate-deuterated cultures were supplemented with 1, 2 or 4 g/L of culture of u-[$^2$H,$^{13}$C,$^{15}$N] labeled ISOGRO®, as well as the standard 2 g/L of $^2$H,$^{13}$C D-glucose and 1



g/L $^{15}NH_4Cl$. Cells were grown at 37 °C until OD (600 nm) reached ca. 0.6-0.8. Protein expression was induced using IPTG (1 mM). Cells were harvested by centrifugation after 3-5 hours of expression, and pellets were resuspended in Lysis buffer M. Lysis was performed by sonication, using a Q700 Ultrasonic Processor (Qsonica), at 40% amplitude and for a total operating time of 6 minutes. Cell lysates were heated to 70 °C for 20 minutes and subsequently centrifuged (40000 rcf, 1 hour, 4 °C). The supernatant was fitered and protein purification was conducted using a Resource Q column (GE Healthcare) equili-

brated in Buffer A, eluting the protein using a linear gradient of Buffer B over 10 column volumes. Fractions were analysed by SDS-PAGE (12.5 % polyacrylamide) and the protein was identified by its monomeric molecular weight (33.55 kDa) and concentrated using an Amicon 10 kDa concentrator (Millipore). The concentrated solution was therefore loaded onto a HiLoad 26/600 Superdex 200pg column (Sigma-Aldrich) equilibrated with Buffer A. MAS rotors were filled for NMR measurements by ultracentrifugation (68300 rcf, over night).

Bacteriophage T5 tail tube protein pb6 modified in C-ter with TEV cleaving site and hexahistidine-tag was produced by overexpression in E. coli BL21(DE3) (SingleTM competent cells Merck) of a *pb6*-pLIM13 plasmid. Cells were grown either (i) from deuterated pre-culture in $D_2O$-based minimal buffer (M9) or (ii) with home-made u-$[^2H,^{13}C,^{15}N]$ cell lysate from a deuterated bacterial culture added at a concentration of 2 g/L to a $H_2O$/M9 culture medium. $^{13}C$-labeled glucose (2 g/L) and $^{15}NH_4$ (1 g/L) were used in the medium in both cases. The purification protocol is reported in ref. Arnaud et al. (2017). Briefly,

cells lysis was performed by 6 passages through a Microfluidizer (M-110 P Microfluidics). Cell lysates were centrifuged for 30 minutes at 3000 rpm (Type 45 Ti Rotor) and 4°C, and pellets were resuspended and incubated for 1 hour at 37°C in Resuspension buffer P. They were then applied to a sucrose cushion gradient (50, 40, 35 and 35% sucrose cushion, Sucrose buffer), and spun for 30 minutes at 30000 rpm (SW41 rotor) and 4°C. The obtained pellet was energetically resuspended in Buffer P and pelleted/resuspended several times by low-speed centrifugation, to remove sucrose contamination.

Ubiquitin samples, prepared to estimate the labelling amount, were expressed in *Escherichia coli* BL21(DE3) cells, transformed with a pET-21b plasmid carrying the human Ubiquitin gene. For the production of perdeuterated ubiquitin, transformants were adapted progressively in four stages over 48 h to M9/$D_2O$ media containing 1 g/L $^{15}ND_4Cl$, 2 g/L $^2H,^{13}C$ D-glucose as the sole nitrogen and carbon sources. In the final culture, the bacteria were grown at 37 °C. When the optical density at 600 nm ($OD_{600}$) reached ca. 0.8-0.9, protein expression was induced by addition of IPTG to a final concentration of 1 mM and

cells were incubated for another 3 h at 37 °C. For the ubiquitin samples prepared with cell lysate (either ISOGRO® or in-house made cell lysate, see below) we used either the same adapatation protocol and a $D_2O$-based preculture, or a shortened protocol without adaption state and with a $H_2O$-based preculture. The $D_2O$ preculture does not have an effect on the final labelling (Fig. 2B). In all cases, cells were harvested by centrifugation, resuspended in 20 mL of Lysis buffer U, and lysed by sonication. The lysate was centrifuged for 30 minutes at 46,000 g (JA25-50 Beckman rotor), and the supernatant was dialyzed against two

times 300 mL of Buffer U. After dialysis the sample was centrifuged for 30 minutes at 46,000 g and loaded on a 40 mL Q-Sepharose column. Ubiquitin was recovered in the flow-through fractions, which were subsequently concentrated and injected on a HiLoad 16/60 Superdex 75 column equilibrated with 1 column volume of Buffer U. The buffer was exchanged to pH 6.5 for solution-NMR.





## 3.2 NMR spectroscopy

MAS NMR spectra of the proteins TET2 (except those of Fig. 3) and pb6 were acquired on a Bruker Avance 3 HD (600 MHz) equipped with a 1.3 mm probe tuned to $^1$H, $^{13}$C, $^{15}$N with an auxiliary coil tuned to $^2$H. The effective sample temperature of MAS NMR experiments was kept at ca. 28 °C, measured from the bulk water frequency, using the relationship T[° C]=255 - 90 · $\delta_{H20}$, where $\delta_{H20}$ is the bulk water Böckmann et al. (2009) frequency in ppm. The chemical shift was referenced with respect to the signal of MPD at 4.1 ppm, which is not significantly dependent on temperature (Fig. S1 of ref. Gauto et al.

(2019b)). Additional spectra of TET2 at 100 kHz MAS frequency (Fig. 6) were recorded on a Bruker Avance 3 HD (950 MHz) spectrometer equipped with a 0.7 mm probe tuned to $^1$H, $^{13}$C, $^{15}$N. Experiments for T$_2$' measurements (Fig. 3) were recorded on a Bruker NEO spectrometer (700 MHz) equipped with a 0.7 mm HCN probe (Fig. 3). Experiments with MalDH were recorded on Bruker NEO spectrometers operating at 600 MHz or 700 MHz as specified in Fig. 4D, E, using 1.9 mm HXY probes tuned to $^1$H, $^{13}$C, $^{15}$N.

All experiments reported herein (2D hNH, 2D hCH, 3D hCANH, 3D hCONH, 4D HCANH) were recorded with pulse sequences available in the NMRlib library Vallet et al. (2020). All transfers in these experiments used cross-polarization transfer steps. The 4D HCANH (Fig. 6A) used the following CP parameters: 40 kHz ($^{13}$C) and 92 kHz ($^1$H, linear ramp 90-100) for the H-CA transfer (4 ms), 12.5 kHz ($^{13}$C) and 42 kHz ($^{15}$N, linear ramp 70-100) for the CA-N transfer (8 ms) and 40 kHz ($^{15}$N) and ca. 90 kHz ($^1$H), linear ramp 90-100) for the N-H transfer (1 ms). The indirect dimensions were as follows: CA

34 ppm, 88 points; N 32 ppm, 120 points; HA 12 ppm, 58 points. The recycle delay was set to 0.65 s, and the total duration was ca. 6 days. Similar CP parameters were used for hCONH, hCANH and hNH and hCH experiments recorded at 55 kHz. Typical experimental times for the 3D experiments were 2 days.

The hNCAHA experiment performed at 100 kHz (950 MHz) used the following CP parameters. H-N CP at 22 kHz ($^{15}$N) and 122 kHz ($^1$H, ramp 90-100; 1 ms), N-CA 38 kHz ($^{15}$N, ramp 90-100, 4 ms) and 70 kHz ($^{13}$C), and CA-HA CP at 30 kHz ($^{13}$C)

and 125 kHz ($^1$H, ramp 90-100, 1 ms); the total experimental time was 2 days. The CP setting used for the hCH experiment was the same as the CA-HA setting above.

The hCANH experiment performed at 38 kHz for MalDH (*600*/700 MHz) used the following CP parameters. H-N CP at *24*/48 kHz ($^{15}$N) and *63*/85 kHz ($^1$H, ramp 90-100; 1 ms) H-CA CP at *23*/50 kHz ($^{13}$C) and *63*/82 kHz ($^1$H, ramp 90-100; 4 ms), N-CA *28*/30 kHz ($^{15}$N, ramp 90-100; 10 ms) and *8*/7 kHz ($^{13}$C). The experimental time was 2 days.

Solution-state NMR of ubiquitin was performed on a 700 MHz Bruker Avance 3 HD spectrometer equipped with a cryoprobe to quantify the amount of deuteration. $^{13}$C-HSQC experiments were recorded on samples of ca. 0.3-0.5 mM concentration with a $^{13}$C spectral width of 70 ppm (512 complex points) using a 3 s recycle delay. The intensity was normalized by the sample concentration (confirmed by 1D $^1$H spectral intensity), and the deuteration level was obtained by comparing the peak intensities of aliphatic signals with those of the fully protonated sample. The results for the Hα sites were confirmed using an alternative

method based on HNcoCA experiments omitting $^1$H decoupling, as done by Löhr *et al.* Löhr et al. (2003).

For all NMR spectra, spectral inspection, manual peak-picking and chemical shifts assignment were performed using the CCPNMR software Vranken et al. (2005). Routines for fits of coherence decays (Fig. A1 were written in python.





## 3.3 Mass spectrometry

The deuteration level of MalDH produced with different culture-media compositions (Fig. 1A) was determined by intact mass
spectrometry at the MS facility of the Max Perutz Labs. Briefly, samples were diluted to 20 ng/μL in $H_2O$, and 30 - 40 ng
were loaded on a XBridge Protein BEH C4 Column, 300 Å, 2.5 $\mu$m, 2.1 mm x 150 mm (Waters), using a Dionex Ultimate
3000 HPLC system (Thermo Scientific). The proteins were eluted with an acetonitril gradient from 12 to 72 % in 0.1 % formic
acid at a flow rate of 250 µL/min. Mass spectra were recorded in the resolution mode on a Synapt G2-Si mass spectrometer
equipped with a ZSpray ESI source (Waters). Glu[1]-Fibrinopeptide B (Glu-Fib) was used as a lock mass, and spectra were
corrected on the fly. Data were analyzed in MassLynx version 4.1 using the MaxEnt 1 process to reconstruct the uncharged
average protein mass.

## 3.4 Preparation of in-house made cell lysate

We prepared a substitute for the commercial algal medium by using unused parts from cultures prepared under perdeuteration
conditions, i.e. from proteins usually considered as contaminants. Standard perdeuteration conditions were used for the cultures
($D_2O$, 2 g/L $^2H$,$^{13}C$ glucose, 1 /L $^{15}N$ ammonium chloride). We prepared such cell lysates from two independent cultures. The
proteins produced in these cultures were either TET2 or ClpP (as reported in ref. Felix et al. (2019)). In both cases, the proteins
are expressed in the soluble fraction (rather than in the pellet of the membrane/inclusion bodies). In one experiment, performed
with ClpP, the protein of interest was purified by Ni-affinity chromatography. All wash fractions from this affinity chromatog-
raphy step (prior to eluting ClpP) with significant amount of protein were pooled and dialysed against water to eliminate the
imidazol contained in the wash steps. The solution containing the contaminant proteins was then acidified with 1 M phosphoric
acid (1 M final) and left at 80 ° in an incubation oven for 5 days to allow for acid-catalysed peptide-bond hydrolysis. The
rationale for using phosphoric acid is that after neutralisation the sample contains phosphate, a buffer component used in M9
media. Thereafter, it was neutralised with NaOH, cleared by centrifugation and lyophilised.

The second approach, performed with ClpP and TET2, used the debris after lysis of the cells and centrifugation; rather than
using the soluble fraction that was taken above, the insoluble fraction was subjected to the same acid hydrolysis treatment
described above. The amino-acid composition from all three samples was very similar.

## 3.5 Amino-acid analysis

For the determination of the amino-acid composition of the in-house made lysate, the lyophilised powder of the samples
described in the previous section, doped with a precisely known amount of norleucine, were dried and then hydrolyzed for 24
hours at 110 °C in 6 N HCl containing 1 % (w/v) phenol. Samples were then dried and resuspended in analysis buffer and
injected into an ion-exchange column of a Biochrom 30 amino acid analyzer, using measurements of the optical density at 570
and 440 nm. Quantification was performed with respect to the internal norleucine standard.



# 4   Conclusions

We have demonstrated the utility of a protein deuteration method that uses $H_2O$ based M9 media enhanced with deuterated
amino-acid mixtures for high-resolution [1]H-detected MAS NMR. While all exchangeable hydrogen sites, and in particular
amides, are protonated with this approach, the deuteration level elsewhere in the protein is high (80 % overall). Compared
to the previously proposed iFD method Medeiros-Silva et al. (2016) our approach enhances the deuteration ca. 4-fold. The
deuteration level is lower for $\alpha$ sites and varies depending on the amino-acid type, presumably due to the efficiencies of the
transaminases acting on different amino acid types. Despite the substantial residual H$\alpha$ protonation, which leads to somewhat
shorter amide-[1]H coherence life times (Fig. 3), the line widths, which include inhomogeneous contributions, are similar to
those of perdeuterated samples.

Our approach is quite similar to the previously proposed method of deuteration in $H_2O$ which uses a bacterial growth
medium (such as SILEX® or ISOGRO®) without the other M9 medium components. Our approach is currently economically
more interesting than a method using only growth media such as SILEX® used by Löhr *et al.*, because the complex media
are more expensive (list price ca. 6000 € for the amount recommended for 1 L of culture) than the components used here.
While isotope prices can vary greatly and evolve over time, we illustrate our rationale using the best prices we obtained from
different providers end of 2023 ($D_2O$ ca. 400 €/L, [2]H,[13]C D-glucose ca. 300-400 €/g, [13]C D-glucose ca. 300 €/g algal lysate
[2]H,[13]C,[15]N ca. 200-350 €/g), which result in ca. 1500 €/L of culture for a sample deuterated in $H_2O$ with cell lysate, and ca.
1200 €/L for a perdeuterated sample. Of course, these numbers are to be taken with caution as they vary.

An additional advantage of the use of cell lysates, compared to perdeuteration, is that they often also boost protein production
significantly, effectively lowering the cost of this approach further, compared to perdeuteration. The details depend very much
on the protein and expression conditions.

Compared to the use of protonated samples used at very high MAS frequencies, the deuteration approach explored here
has significant advantages: line widths are better for amide-[1]H signals in deuterated samples at frequencies achieved with 1.3
mm rotors compared to those obtained with protonated samples spun at 100 kHz (Fig. 3C), and it comes with a significant
sensitivity gain. In our hands, experiments on 1.3 mm (deuterated protein) vs 0.7 mm (protonated protein) samples show the
sensitivity gain estimate of a factor 2.5, proposed by Le Marchand *et al.* Le Marchand et al. (2022) is a lower-limit estimate.
Thus, deuteration, while providing better resolution, is also very beneficial in terms of sensitivity.

Our approach can of course also be used with a $H_2O$/$D_2O$ mixture, which will result in even higher deuteration levels. For
example, using 50 % $D_2O$ in the growth medium will result in 50 % protonation of the non-exchangeable sites and 100 %
for the exchangeable sites (provided the final sample is in $H_2O$). Using this approach enables the detection of all amide sites
without lengthy exploration of refolding protocols with unsure outcome. In new projects in our group we apply this approach
right from the start whenever the involved protein is likely to have non-exchangeable amides, thereby saving precious time and
ensuring high quality samples.

We are aware of a very recently deposited manuscript on bioRxiv that describes a similar approach, namely the use of 10 g/L
ISOGRO® without M9 medium components for deuteration in $H_2O$ Aucharova et al. (2024), although at significantly higher



costs that what we propose here. While that study has a more limited analysis and does not provide details of the additional assignment it enabled, it is interesting that similar conclusions are drawn as in our study. Those results further corroborate our data and support that this kind of deuteration approach may become a standard method for large proteins.





**Appendix A**

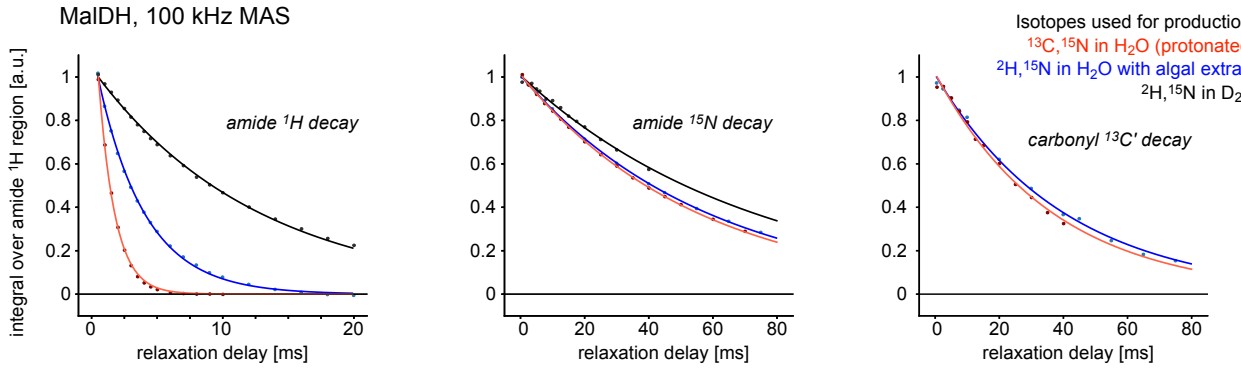

**Figure A1. Examples of coherence decays in differently labelled samples.** The decay curves correspond to the data marked with an asterisk in Figure 3A. The solid lines are fits to the data, using a mono-exponential decay function. The data have been normalised such that the fitted curve at a relaxation delay of zero is at unity.

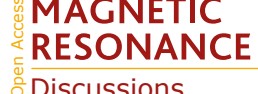

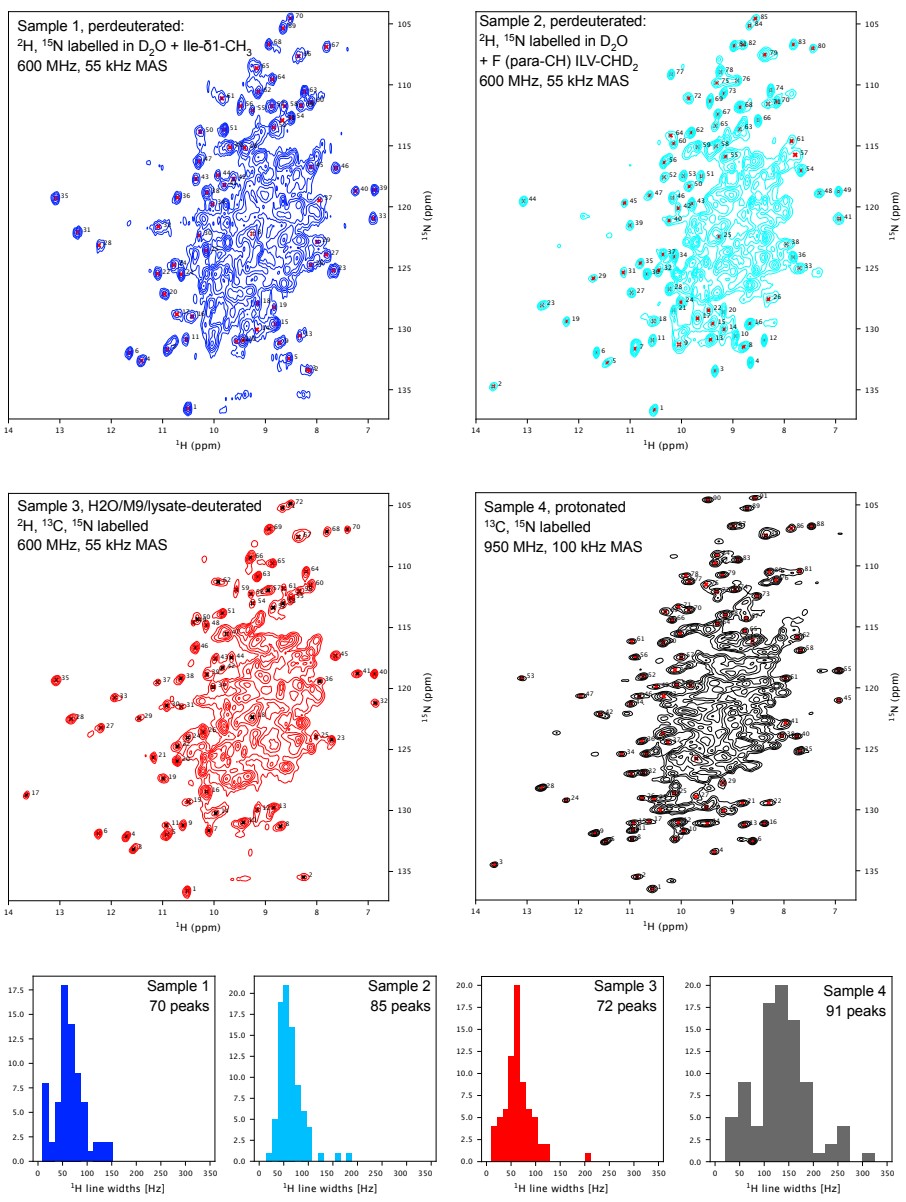

**Figure A2. Line widths in differently labelled samples.** These spectra have been used to extract the $^1$H line width statistics reported in Figure 3C. Samples 1 and 2 were deuterated in a $D_2O$ based culture; they have been used for previous studiesKurauskas et al. (2016); Gauto et al. (2019a), for which additional labelling has been used as indicated. Note that this labelling introduces protons only at the terminal methyl of Ile (sample 1) or a single hydrogen at the terminal methyls of Ile, Leu, Val and the ζ-position of Phe (sample 2), far from the amide sites, and does not lead to significant changes of amide $^1$H line widths. Sample 3 has been deuterated in $H_2O$ based medium supplemented with deuterated ISOGRO®. Sample 4 is fully protonated; note that the latter has been measured at 950 MHz $^1$H Larmor frequency, while all other spectra were measured at 600 MHz. Crosses indicate the peaks that have been used for the analysis, and the plots at the bottom indicate the binned line width distribution.



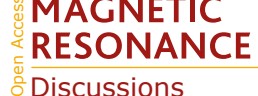

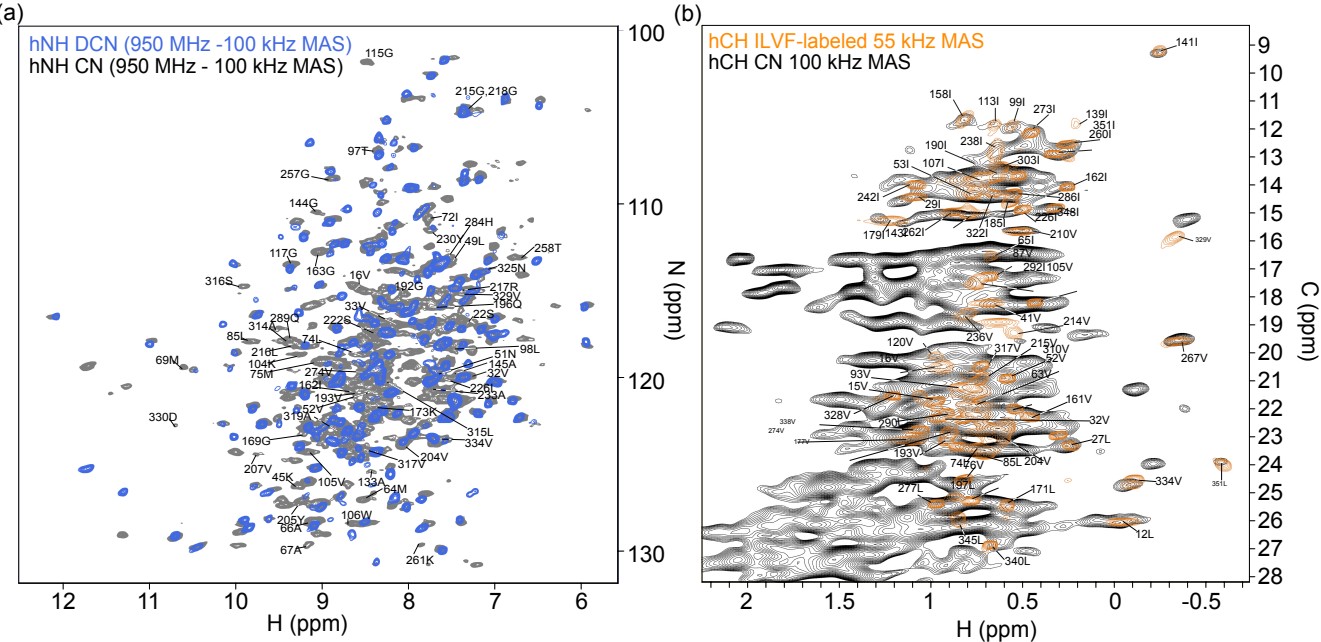

**Figure A3. Ultra-fast MAS of a fully protonated and deuterated TET2 samples indicate the usefulness of deuteration.** (a) 2D hNH overlay of perdeuterated ($^1$H,$^{13}$C,$^{15}$N; final sample in $H_2O$; blue) and protonated ($^{13}$C,$^{15}$N) TET2 samples. Newly assigned amino-protons are labeled. (b) Overlay of 2D hCH spectra of the protonated sample (black) with a sample that is $^{13}$CHD$_2$ labeled at Ile ($\delta$1), Leu ($\delta$1) and Val ($\gamma$1) as well as the *para*-CH site of Phe Gauto et al. (2019b). The resolution gain with the specifically methyl-labelled sampled is obvious.

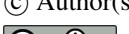



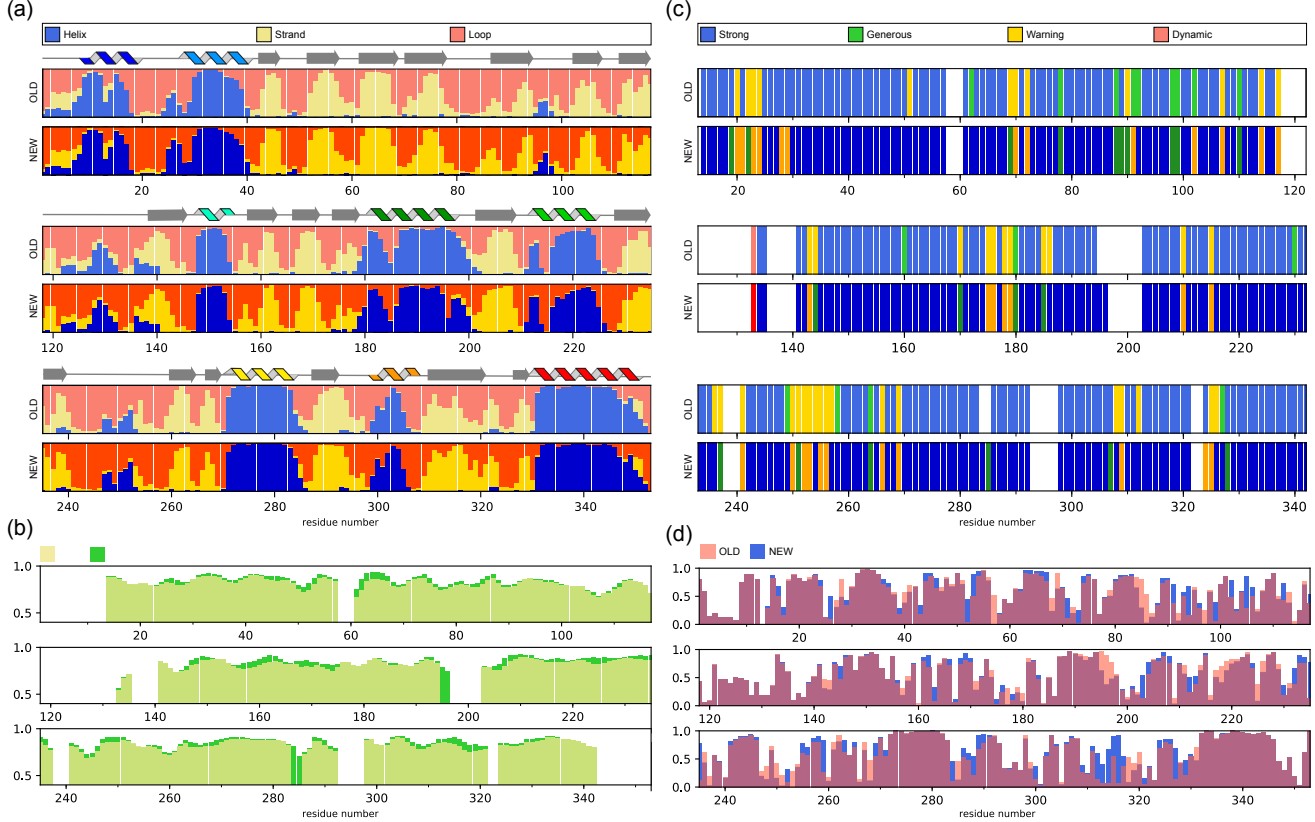

**Figure A4. TALOS-N analysis of the previously-deposited (BMRB 27211) and newly-determined chemical shift assignments.** (a) TALOS-N prediction of the secondary-structure (helix, strand or loop) probabilities per residue. The predictions for the two data sources is almost identical, and both are in agreement with the secondary-structure elements from the crystal structure (printed over the bar graphs). (b) Predicted RCI-S$^2$ backbone order parameters highlight an overall decrease of the protein flexibility and improvement of the assignment self-consistency. (c) TALOS-N evaluation of the assignment's quality per residue. The new data show an improvement in Strong (228→238) and Generous (16→20) assignments, and a decrease of Warning ones (41→31). (d) TALOS-N confidence of the 3-state (helix, sheet or coil) secondary-structure prediction per residue.



**Table A1.** Residues for which the chemical shifts of amino- and α-protons were assigned. Residues that could not be assigned in the sample above are highlighted in bold.

| Labeling | Assignment |
|---|---|
| DCN | H: 14, 17, 18, 20, 21, 23, 24, 25, 26, 28, 29, 35, 36, 38, 39, 41, 42, 43, 44, 46, 47, 48, 50, 55, 56, 57, 62, 70, 71, 73, 78, 79, 80, 81, 82, 83, 84, 87, 90, 91, 92, 93, 94, 96, 100, 101, 102, 108, 109, 110, 111, 112, 113, 114, 118, 135, 146, 147, 148, 149, 150, 152, 153, 155, 157, 158, 159, 161, 165, 166, 168, 170, 171, 172, 174, 179, 181, 182, 184, 185, 186, 195, 203, 210, 211, 214, 219, 221, 223, 224, 225, 227, 229, 231, 235, 236, 238, 239, 240, 241, 242, 244, 245, 247, 248, 249, 251, 252, 253, 255, 262, 263, 264, 265, 269, 270, 272, 273, 275, 276, 283, 288, 290, 291, 292, 298, 301, 302, 303, 304, 305, 306, 307, 308, 309, 310, 312, 313, 321, 326 |
| ISOGRO® | H: 14, 17, 18, 20, 21, 23, 24, 25, 26, 28, 29, 35, 36, 38, 39, 41, 42, 43, 44, 46, 47, 48, 50, 55, 56, 57, 62, 70, 71, 73, 78, 79, 80, 81, 82, 83, 84, 87, 90, 91, 92, 93, 94, 96, 97, 100, 101, 102, 108, 109, 110, 111, 112, 113, 114, 118, 135, 146, 147, 148, 149, 150, 152, 153, 155, 157, 158, 159, 161, 165, 166, 168, 170, 171, 172, 174, 179, 181, 182, 184, 185, 186, 195, 203, 210, 211, 214, 219, 221, 223, 224, 225, 227, 229, 231, 235, 236, 238, 239, 240, 241, 242, 244, 245, 247, 248, 249, 251, 252, 253, 255, 262, 263, 264, 265, 269, 270, 272, 273, 275, 276, 283, 288, 290, 291, 292, 298, 301, 302, 303, 304, 305, 306, 307, 308, 309, 310, 312, 313, 321, 326, **16, 22, 27, 32, 33, 45, 49, 51, 52, 54, 64, 66, 67, 69, 74, 75, 85, 98, 104, 105, 106, 115, 117, 133, 144, 145, 162, 163, 169, 173, 192, 193, 196, 204, 205, 207, 215, 216, 217, 222, 226, 230, 233, 257, 258, 261, 274, 284, 289, 314, 315, 316, 317, 319, 325, 329, 330, 334** <br><br>HA: **16, 18, 20, 22, 25, 26, 27, 28, 32, 33, 35, 38, 41, 45, 46, 47, 49, 50, 51, 52, 53, 56, 57, 62, 63, 64, 65, 66, 67, 69, 70, 73, 74, 75, 78, 79, 81, 82, 84, 85, 87, 90, 91, 92, 93, 94, 96, 98, 101, 102, 105, 106, 110, 111, 113, 115, 117, 133, 135, 144, 145, 146, 147, 148, 149, 150, 157, 158, 159, 161, 162, 163, 165, 166, 168, 169, 171, 173, 174, 179, 181, 185, 192, 193, 196, 203, 204, 205, 207, 210, 211, 212, 214, 215, 216, 221, 222, 224, 225, 226, 230, 231, 233, 235, 236, 241, 242, 244, 245, 247, 248, 250, 251, 252, 253, 255, 257, 258, 260, 262, 263, 264, 265, 269, 272, 273, 274, 276, 279, 284, 288, 289, 290, 298, 301, 303, 304, 305, 306, 307, 308, 309, 310, 312, 313, 314, 315, 316, 317, 319, 321, 325, 326, 329, 330, 331, 334** |
| CN | H: 14, 17, 18, 20, 21, 23, 24, 25, 26, 28, 29, 35, 36, 38, 39, 41, 42, 43, 44, 46, 47, 48, 50, 55, 56, 57, 62, 70, 71, 73, 78, 79, 80, 81, 82, 83, 84, 87, 90, 91, 92, 93, 94, 96, 100, 101, 102, 108, 109, 110, 111, 112, 113, 114, 118, 135, 146, 147, 148, 149, 150, 152, 153, 155, 157, 158, 159, 161, 165, 166, 168, 170, 171, 172, 174, 179, 181, 182, 184, 185, 186, 192, 195, 203, 210, 211, 214, 219, 221, 223, 224, 225, 227, 229, 231, 235, 236, 238, 239, 240, 241, 242, 244, 245, 247, 248, 249, 251, 252, 253, 255, 262, 263, 264, 265, 269, 270, 272, 273, 275, 276, 283, 288, 290, 291, 292, 298, 301, 302, 303, 304, 305, 306, 307, 308, 309, 310, 312, 313, 321, 326, 16, 22, 27, 32, 33, 45, 49, 51, 52, 54, 64, 66, 67, 69, 74, 75, 85, 97, 98, 104, 105, 106, 115, 117, 133, 144, 145, 162, 163, 169, 173, 192, 193, 196, 204, 205, 207, 215, 216, 217, 222, 226, 230, 233, 257, 258, 261, 274, 284, 289, 314, 315, 316, 317, 319, 325, 329, 330, 334, **72, 208, 218** <br><br>HA: 16, 18, 20, 22, 25, 26, 27, 28, 32, 33, 35, 38, 41, 45, 46, 47, 49, 50, 51, 52, 53, 56, 57, 62, 63, 64, 65, 66, 67, 69, 70, 73, 74, 75, 78, 79, 81, 82, 84, 85, 87, 90, 91, 92, 93, 94, 96, 98, 101, 102, 105, 106, 110, 111, 113, 115, 117, 133, 135, 144, 145, 146, 147, 148, 149, 150, 157, 158, 159, 161, 162, 163, 165, 166, 168, 169, 171, 173, 174, 179, 181, 185, 192, 193, 196, 203, 204, 205, 207, 210, 211, 212, 214, 215, 216, 221, 222, 224, 225, 226, 230, 231, 233, 235, 236, 241, 242, 244, 245, 247, 248, 250, 251, 252, 253, 255, 257, 258, 260, 262, 263, 264, 265, 269, 272, 273, 274, 276, 279, 284, 288, 289, 290, 298, 301, 303, 304, 305, 306, 307, 308, 309, 310, 312, 313, 314, 315, 316, 317, 319, 321, 325, 326, 329, 330, 331, 334, **72, 208, 218, 43, 54, 97, 104, 170, 217, 249, 261** |



**Table A2.** Buffers used during sample preparations

| Protein, Buffer | Composition |
|---|---|
| TET2, Lysis buffer T | 50 mM Tris, 150 mM NaCl, 20 mM MgSO$_4$, 0.1 % Triton X-100, 0.025 mg/ml lysozyme, 0.05 mg/ml deoxyribonuclease, 0.2 mg/ml ribonuclease (pH 8) |
| TET2, Dialysis buffer | 20 mM Tris, 100 mM NaCl (pH 7.5) |
| TET2, Elution buffer | 20 mM Tris, 1 M NaCl (pH 8) |
| TET2, NMR buffer | 20 mM Tris, 20 mM NaCl (100 % H$_2$O, pH 7.6) |
| MalDH, Lysis buffer M | 50 mM Tris, 50 mM NaCl, 2 mM MgCl$_2$, 0.05 mg/ml deoxyribonuclease, 0.25 mg/ml ribonuclease, cOmplete EDTA-free protease inhibitor (pH 7) |
| MalDH, Buffer A | 50 mM Tris, 50 mM NaCl (pH 7) |
| MalDH, Buffer B | 20 mM Tris, 1 M NaCl (pH 7.5) |
| Ubiquitin, Buffer U | 50 mM Tris (pH 8) |
| Ubiquitin, Lysis buffer U | 50 mM Tris, 2 µg/mL leupeptine, 2 µg/mL pepstatine (pH 8) |
| pb6, Resusp. buffer P | 50 mM Tris, 100 mM NaCl, 0,5% Triton X-100, 1 mM EDTA, 100 µg/ml lysozyme (pH 8) |
| pb6, Sucrose buffer | 20 mM Tris, 100 mM NaCl, 0,05% Triton X-100 (pH 8) |
| pb6, Buffer P | 20 mM Tris, 100 mM NaCl (pH 6.9) |



*Code and data availability.*  All pulse sequences are available in the NMRlib library and from the authors. Analysis scripts for coherence-decay experiments and line widths analysis are available from the authors. The new resonance assignments are submitted to the BioMagRes-Bank (https://bmrb.io/; accession number pending). Other data are available from the authors upon request.

*Author contributions.*  F. N. prepared all MalDH samples, and TET samples for $T_2$' measurements, recorded and analysed all $T_2$' and line
width data, recorded all multidimensional experiments of MalDH, performed TET2 resonance assignment and TALOS analysis. J.-Y. G. prepared ubiquitin samples and analysed the labelling pattern, and a TET2 sample. C.-A. A. and C. B. prepared pb6 samples. P. M. and H. F. prepared cell lysates from bacterial cultures. P. S. designed the study and recorded NMR experiments. F. N. and P. S. prepared all figures and wrote the manuscript.

*Competing interests.*  At least one of the (co-)authors is a member of the editorial board of Magnetic Resonance.

*Acknowledgements.*  We thank Dominique Madern (IBS Grenoble) for providing the plasmid for MalDH and feedback on the manuscript, Alicia Vallet for excellent support at the Grenoble NMR facility and Petra Rovo and Margarita Valhondo at the IST Austria NMR Service Unit. We thank Dorothea Anrather in the Mass Spectrometry Facility of the Max Perutz Labs for the mass spectrometry analysis using the instruments of the Vienna BioCenter Core Facilities (VBCF). We are grateful to Jean-Pierre Andrieu (Plateforme Seq3A, IBS Grenoble) for the analysis of the amino-acid composition of the in-house made lysates. Part of this work used the platforms of the Grenoble Instruct-ERIC
center (ISBG ; UAR 3518 CNRS-CEA-UGA-EMBL) within the Grenoble Partnership for Structural Biology (PSB), supported by FRISBI (ANR-10-INBS-0005-02) and GRAL, financed within the University Grenoble Alpes graduate school (Ecoles Universitaires de Recherche) CBH-EUR-GS (ANR-17-EURE-0003). IBS acknowledges integration into the Interdisciplinary Research Institute of Grenoble (IRIG, CEA). C.-A. A. was funded by GRAL. We are grateful to Rasmus Linser (Univ. Dortmund) for sharing a paper draft describing a similar study. This work was supported by the Austrian Science Fund (FWF; project number I5812-B).



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
