# Peer review of "Deuteration of proteins boosted by cell lysates: high-resolution amide and $H\alpha$ MAS NMR without re-protonation bottleneck"

_Magnetic Resonance, 2024_

## Author Response (AR1)

**CC:**
*Dear Authors*
*First of all: very nice work. Very useful particularly for samples that are difficult to back exchange.*
*You focus on MAS NMR but this could also be useful for solution NMR of larger proteins - couldn't it?*

We focused on MAS NMR, and indeed the approach also work for solution-state NMR. Besides the publication from Löhr and co-workers, we have now added a more recent paper (O'Brien 2018) where the authors also use Isogro for labelling, and apply it to solution-state NMR.

*In Figure 1 b/c you analysed in detail the side chain protonation level for Ubiquitin produced with algal extract. It would be nice to see a figure like this also for a sample made with your homemade Ecoli extract.*

We have not performed the analysis of the deuteration levels at the same level of detail for both types of sample and at this point we can only show the 2D spectra comparison. See the reply to Reviewer 2.

*distance restraints: In fully protonated samples and fast-MAS it is difficult to obtain a high number of 1H-1H distance restraints due to "dipolar truncation" and the absence of methods to efficiently overcome this issue. Maybe you could investigate how your labeling scheme is helping with that. For example you could compare a hCHH RFDR of fully protonated Ubi and partially deuterated Ubi (maybe with 2H decoupling if you have a 4 channel or an HXY probe).*

Distance restraints: this is a very good idea. It has not been our focus here to go into structure restraints, and unfortunately we do not have any samples any more where we could do this in a comparable manner. As it would be a pretty significant effort to make these samples and record and analyse these spectra, we prefer not to go in this direction at this point and rather publish for now the principle and performances of the labelling scheme.
The proposed experiment, a RFDR on a partially deuterated sample, will have a potential gain, as pointed out by your comment: one might see longer-range contacts. On the other hand, the remaining protons are only there to a certain level: looking at figure 1b one can see that in the side chain, only 10-40% are protonated. Thus, cross-peak intensities will be scaled down by this level squared. This may be prohibitively low. On the other hand, one may likely be able to do such an experiment with a 1.9 mm rotor, thus gaining a lot of sensitivity from the sample amount.
The better way is certainly to do a proper 100% labelling of e.g. methyl groups and have all the rest deuterated; this has been done by several groups.
We leave this question for later experiments, i.e. we have not included anything into the current paper.

**RC1**:

*Napoli et al. present the use of deuterated cell lysate for solid-state NMR sample production. The authors produced the deuterated cell lysate from fractions of perdeuterated cultures. These fractions are normally thrown away, and therefore it's a clever approach to keep these fractions and use them again for future expressions. The authors carefully analyse the effect of their labeling scheme on the NMR spectral quality which turns out to be very promising. I thus support publication of this method, which is for sure a very nice addition to the NMR labeling tool box.*

We thank the reviewer for the careful evaluation of the manuscript.
Here we address the four points:

*- I would encourage the authors to compare in the conclusion section better (and maybe fairer?) their approach against the 0.7 mm / fully protonated approach: Costs of rotors, fragility of the equipment, but also required sample amount, increasing prices for deuterated glucose etc.*

We have now provided a paragraph in the Conclusions section that describes the costs of the samples, as well as rotors.

*- The authors should discuss the possible application of their approach towards proton-detected ssNMR studies of membrane proteins*

We have added a sentence in the Conclusions part about the potential use of the approach for membrane proteins.

*- One slight drawback of the method is that one still needs to add the standard 2 g/L of deuterated glucose in addition to the amino acid mix from the cell lysates. Would it be possible to reduce the amount of deuterated glucose that needs to be added?*

This is a very good point, and yes, we could likely reduce the amount of glucose further. A study by O'Brien et al has used 1 g/L unlabeled glucose and reported similar deuteration levels. We have discussed this possibility now in the Conclusions section. We have done the vast majority of our study before the O'Brien paper was published.

*- The optimum labeling method will vary from system to system and depend on external parameters as well including MAS rate and magnetic field strength. I believe the introduced method could be quite useful in many cases but it may not be the optimum solution in all cases. I think if the commercially available 0.7 mm probes would improve in reliability and the prices for 0.7 mm rotors would drop this methodology could become quite powerful and wide-spread.*

Yes, the choice of the method depends certainly on the circumstances (such as probe sensitivity, and even the inherent line width of the protein). From our own experience, as well as from data reported by others (e.g. the Chem. Rev. review by the Pintacuda group which we cited), it is clear that in terms of sensitivity a 0.7 mm probe is significantly worse than a 1.3 or 1.9 mm probe. As the line widths resulting from 100 kHz MAS on a protonated sample

and 40 kHz MAS from our partially deuterated samples are comparable (Figure 3), it is difficult to argue why the 0.7 mm probe would be preferable, in our opinion.

We think that not the price of the 0.7 mm probe would be the game changer; what would really make a significant difference is if 0.7 mm probes gained a factor of 2-3 in sensitivity. We have added a short discussion in the Conclusions section, including an estimate of the prices.

**RC2:**

*The paper of Napoli et al. describes a very interesting work on the use of cell lysates for the preparation of isotopically labeled proteins for NMR studies. This is a very interesting approach, in particular nowadays where the costs for isotopic labeled nutrients are exploding.*

We thank the reviewer for the positive evaluation and the additional and very valid points. We reply below to these points.

*1) An alternative strategy to yield incorporation of amide protons and random protonation in protein side chains was suggested by Asami et al. - There, bacteria are grown in a medium containing 2H,13C glucose and various amounts of H2O. These authors have also suggested an assignment strategy to yield the chemical shifts of aliphatic protons. I am wondering whether the approach presented here (which uses 100% H2O) yields a higher sensitivity in amides / alpha sites. It would be interesting to add this to the discussion of this paper.*

We have added a brief discussion of the approach in the introduction section as well as in the Conclusions section. We were aware of the approach (and one of us co-authored one of the papers), and although amide-detection was not the goal of the "RAP" approach, it is indeed interesting to mention it.
We have now started the Conclusions section with a brief general description, which may help some readers get an overview of the various approaches.

*2) The home made cell lysate samples (Fig. 2b) show clear improvements in the 1H spectra for the methyl spectral region (in comparison to the sample prepared using ISOGRO). Please show as well aliphatic/methyl correlation spectra for the three samples, and add them at least to the appendix.*

This is indeed an interesting point. We have now looked into a series of samples that have been produced during this project, with different batches of bacterial lysate (or even samples from the same batch of lysate). There is a significant variation in the 1D spectra of those samples, and the variation between different samples is as large as the variation between the ISOGRO-labelled sample and the bacterial-lysate-labelled sample of Figure 2b. In other words, the variation that is seen in Figure 2b is likely not relevant. Note that the residual protonation is anyhow pretty low: Figure 1b shows that the methyls are labelled to only a few percent. Thus, even if this protonation level fluctuates by a factor 2, it would not be a major effect (at least for the amides).
Overall, these data have convinced us that the variations are actually rather small, and we would not want to draw conclusions whether ISOGRO vs bacterial-lysate deuteration results in a different deuteration pattern.
We have added statements in the description of Figure 2b which makes clear that the fluctuations are not really relevant.

*3) The incorporation of protons at alpha sites varies strongly depending on amino acid type, and ranges 10% for lysine to 90% in Phe (Fig. 1). Can this be employed for amino acid editing ? How does differential labeling affect the suggested H-alpha assignment experiments presented in 2.6 ? Please discuss.*

We have added a paragraph to the Conclusions section about the possibility to do editing. It is an interesting idea, but probably also complicated by the fact that coherence transfer efficiencies often differ significantly. For example, in crystalline ubiquitin we find a factor of 10 between different amides (e.g. Figure S4 of 10.1002/anie.200904411). We have not found a very clear correlation between the observed HA signals and the type of amino acids – our analysis was certainly not the most comprehensive possible, admittedly.

We find this point interesting, but would like to defer this to a later dedicated analysis, and focus here on the description of the HN detection.

*Met, Trp and Tyr are missing in the plot. It should be mentioned why.*

We have added the following statement: "Amino acid types missing in this plot are either not present in ubiquitin, excluding the possibility for quantification (Trp) or are not visible or unresolved (Met, Tyr)."

*Minor issues:*
*- Formatting of references in the text looks strange. The paranthesis should be around the whole citation and not only around the year of publication.*
*- Figures and captions: panels in figures are labeled using small letters, while capital letters are employed in the caption to refer to the figure. Please homogenize.*
*- p.4, line 123: (u-[2H,13C,15N])*

Thank you for spotting these problems, which we have addressed.